# A Semi-Analytical Method for Designing a Runner System of a Multi-Cavity Mold for Injection Molding

**DOI:** 10.3390/polym14245442

**Published:** 2022-12-12

**Authors:** Chung-Chih Lin, Tian-Cheng Wu, Yu-Shiang Chen, Bo-Yu Yang

**Affiliations:** Department of Mechanical and Computer-Aided Engineering, National Formosa University, No. 64, Wen-Hwa Road, Hu-Wei Town, Yunlin 632301, Taiwan

**Keywords:** multi-cavity mold, rheology, runner balancing design, injection molding

## Abstract

Multi-cavity mold design is an efficient approach to achieving mass production and is frequently used in plastic injection applications. The runner system of a multi-cavity mold delivers molten plastic to each cavity evenly and makes the molded product from each individual cavity possess an equivalent quality. Not only the dimensions, but also the invisible quality, e.g., the internal stress of the product is of great concern in regard to molding quality. Using commercial software to find an optimal solution for the runner system may be time-consuming in respect to iterations if the engineers lack empirical rules. The H-type runner system is often used due to an inherently balanced filling in multi-cavities. However, the shear heat inducing an imbalanced flow behavior requires the H-type runner system to be improved as the number of the cavities is increased. This work develops a methodology based on the rheological concept to determine the runner system of a multi-cavity mold semi-analytically. As the relation of the viscosity with respect to shear rate is known, the runner system can be constructed step-by-step via this method. The use of the proposed method helps to focus attention on the connection between the physical situation and its related mathematical model. The influences of the melt temperature and resin type can be easily investigated. Three design examples, a 16-cavity mold with a fishbone runner system, an 8-cavity mold with an arbitrary runner layout, and the influences of melt temperature and resin type on the runner design are demonstrated and validated by the commercial software. The proposed method shows its great benefit when a new runner design project is launched in the initial design stage and then cooperates with the commercial software for further modifications.

## 1. Introduction

Plastic molding techniques provide a flexible and precise approach to mass production. The advantages of the plastic injection molding, including lightweight, easy modification of the mechanical properties, and efficient production, have made plastic material ever more popular in modern industry. The mold is a key factor for the injection molding process. When the molten plastic is filled into a mold, a desired product, including surface texture, shape and dimension accuracy is formed [1,2,3]. Moreover, the mold also influences the invisible quality, e.g., the internal stress and end-use performance of the molded product [4,5,6]. The functioning of a runner system, which is a material delivery system and usually consists of a sprue, runners (transverse or longitudinal), and gates, is an important factor in achieving successful molding. A sprue connects the nozzle of the injection machine to the mold and an adequate dimension of the nozzle orifice prevents the pressure drop becoming too large. The runner provides a channel for the molten resin to flow forward into the mold cavity. The gate is the thinnest part in the runner system, and therefore, a quick freeze-off of the gate prevents pressurized material in the cavity from returning. For a multi-cavity mold, the runner design exhibits its importance on part quality and molding efficiency. Huang et al. [4] applied four modified symmetrical runner systems to investigate their effect on improving low balance experimentally. A symmetrical runner layout, e.g., the H-type runner, providing an equal path length for each runner, is a well-known approach of a multi-cavity mold design. However, shear heating resulting from the H-type runner makes the melt flow in the symmetrical runner in an imbalanced way and many problems appear, e.g., unstable part dimension, high internal stress, etc. [7]. The melt-flipper technique [8,9] developed by Beaumont et al. reduces the shear heating effect and improves the imbalanced flow problem. However, much more waste volume of the H-type runner system also increases costs associated with regrind with the increase of the number of the mold cavity and reduces its competitiveness. In addition, too many turns designed in the H-type runner system also induce a melt temperature difference at each branch of the runners and causes an unbalanced flow type of the runner system [10]. A family mold usually seen in the industry has more than one cavity cut into the mold, allowing multiple various parts with the same material to be formed in a single cycle. Those parts have some specific requirements, for example, those parts will be assembled into one product. Family molds can often enjoy a mold building cost advantage over molds dedicated to a single part mold, such as easy management of assembly components, less chromatic aberration, etc. Since the parts often have different shapes, the even flow requirement is significantly important for family molds in the melt filling stage. Otherwise, it can lead to an increase in molding defects more frequently than a general multi-cavity mold.

Alam and Kamal used a robust optimization algorithm to determine the optimal runner diameter [11]. A suitable objective function needs to be defined in advance; otherwise, the result will be unacceptable. The design of the experiment also provides an optimal arrangement solution to a balanced flow type [12,13,14]. For the past decades, most of the runner system designs of an injection mold were implemented by the empirical rule or trial and error approach. If a new project is launched and no information can be referred, it makes this design process tedious and costly. Using commercial software is another way to execute the runner system design [15,16,17]. Without an engineering background, the design and analysis process become a series of trial and error operations and wastes much computation cost. In addition, this process lacks a physical essence for the melt flow behavior and needs a lot of empirical practice to improve the efficiency to reach the optimal solution.

In this study, we develop a methodology to design the runner system of the multi-cavity mold based on the rheological concept. As the shear rate with respect to the melt viscosity is known, either by theoretical method or experimentation, the proposed method can be used to determine the diameter of each runner sequentially. With any runners branching from the same junction, the filling time of the molten plastic from the junction to the end of the filling and the pressure drop of the runner must be the same as each other. Thus, the optimal runner system can execute a balanced melt flow of the injection mold. A 16-cavity mold with a fishbone runner system and an arbitrary runner layout of a multi-cavity mold for an even flow are implemented and validated herein. The influence of the melt temperature and the resin type on the runner design is also investigated. 

## 2. Rheological Behavior of Molten Plastic

The flow process of the molten plastic is usually assumed to be a quasi-steady flow of generalized Newtonian uncompressible fluid under a non-isothermal condition [18]. One of the common models used to describe the rheological behavior of the melt is the power law model [18,19]. Although this model is obtained from empiricism, it provides an adequate definition of the non-Newtonian viscosity over the range of shear rates, especially for high shear rates that could be developed during the filling stage of the injection process. For a generalized Newtonian fluid, the relation of the viscosity *η* and the shear rate $\dot{\gamma }$ can be expressed as follows [18]:(1)$$\eta =m{\dot{\gamma }}^{s-1}$$
where *m* is the flow consistency index, and *s* denotes the power law exponent. The viscosity of plastic melts, which is usually modeled as a non-Newtonian fluid, changes with the shear rate. The shearing force in the melt flow process causes an orientation of the molten plastic material parallel to the direction of the applied strain. As the polymer chains become more oriented, or aligned, the steric hindrance between the polymers is less; thus, the molten plastic can move more freely relatively. As a result, the viscosity of molten plastic will decrease with the increasing flow velocity.

For a plastic melt flowing in a cylindrical channel, it is assumed to be a laminar and fully developed flow of the non-Newtonian fluid, as shown in Figure 1. The velocity terms, $\upsilon_{\theta }$ and $\upsilon_{r}$ in the *r*- and θ-directions are in the steady state and equal zero. The term of the derivative operator $\frac{\partial }{\partial \theta }$ in the θ-direction is also zero under the symmetrical condition [18,19]. Hence, the three momentum equations of the plastic melt in the cylindrical coordinate (*r*, θ, *z*) can be simplified as: (2)$$-\frac{\partial P}{\partial r}=0$$
(3)$$-\frac{1}{r}\frac{\partial P}{\partial \theta }=0$$
(4)$$-\frac{\partial P}{\partial z}+\frac{1}{r}\frac{\partial }{\partial r}(r\tau_{rz})=0$$
where *p* and $\tau_{rz}$ are the pressure and the shear stress, respectively. To get the flow velocity of the fluid $v_{z}\left(r\right)$ in a cylindrical channel, the calculation procedures starting from Equations (2)–(4) can be found in references [7,14], and will not be reiterated here for compendiousness. Finally, the $v_{z}\left(r\right)$ can be expressed by: (5)$$v_{z}\left(r\right)=-\frac{sR}{s+1}{\left[\frac{R\Delta P}{2mL}\right]}^{1/s}\left[1-{\left(\frac{r}{R}\right)}^{\left(s+1\right)/s}\right]$$
where Δ*P* is defined as the pressure drop along the flow length *L* and *R* is the runner radius. The negative sign exists because the pressure drop ΔP is negative in an injection molding process, the pressure change can also be regarded as the required pressure when the molten plastic flows along a designated path. Therefore, the pressure drop will be used instead of pressure change ΔP to fit the injection condition. As the velocity is determined, the shear rate $\dot{\gamma }$ is calculated by:(6)$$\dot{\gamma }=\frac{v_{z}}{R}$$

The volume flow rate *Q* in a cylindrical channel can be calculated by taking double integrals of flow velocity of the fluid $v_{z}\left(r\right)$ with respect to the cross-sectional area and expressed as:(7)$$Q=\int_{0}^{2\pi }\int_{0}^{R}v_{z}rdrd\theta =\frac{s\pi R^{3}}{3s+1}{\left(\frac{R\Delta P}{2mL}\right)}^{1/s}$$

Equation (7) expresses that the volume rate of flow is influenced by the geometry of the flow channel, the viscosity of the material, and the pressure drop. For most plastic material modeled as non-Newtonian fluid, the power law exponent *s* is less than 1; thus, the volume rate of flow changes exponentially with respect to the ratio of the required pressure to the viscosity.

The runner system of a mold delivers and distributes the molten plastic from the injection machine to the mold cavity. A balanced design of the runner system can guarantee the product with the same molding quality. Compared to those cross-sectional shapes of the runner system, such as, trapezoid, semi-circle, etc., the circular shape of the runner system is the best due to the cross-sectional area being the biggest under the same circumference. Therefore, we chose the circular shape of the runner system as the model to establish the methodology for calculating the optimal diameter of the runner. The pressure drop Δ*P* can also be regarded as the required injection pressure to resist the friction when the plastic melt flows in the mold during the filling stage and expressed by:(8)$$\Delta P=\frac{2mL}{R}{\left(\frac{Q\left(3s+1\right)}{s\pi R^{3}}\right)}^{s}$$

Equation (8) indicates that the pressure drop is influenced by the molding parameters, e.g., injection velocity, melt viscosity, and the runner’s geometry itself.

## 3. Optimization Methodology

A runner is a channel machined on the parting plane of a mold that allows molten plastic to flow from the nozzle to the cavity. In a variety of runner shapes, e.g., parabolic or trapezoid shape, the full round is the most efficient runner shape because of the lowest pressure drop over the same volume of molten plastic. Therefore, the full round runner was selected as the runner system of a multi-cavity mold for optimization. Assume a multi-cavity mold containing 4*n* cavities with a fishbone runner system, as shown by Figure 2a. In general, the layout of the fishbone runner system and the mold cavity is a symmetrical arrangement; so, the molten plastic filling in the mold condition, no matter whether a balanced or imbalanced filling pattern, is always symmetrical. Assume that the influences of gravity and machining error are tiny. Only half the layout of the runner system (2*n* cavities) is used to demonstrate the optimization for the sake of saving computation time. The runner system of the mold consists of sprue, the transverse runner *T_i_*, and longitudinal runner *L_i_* (*i* = 1, 2, …, *n*), respectively. A concise expression of the runner variable will be used in the following section, e.g., the runner *T_i_* instead of the transverse runner *T_i_* and the runner *L_i_* instead of the longitudinal runner *L_i_*. All of the runners differ in both length and diameter. The symbols, *k_i_*, *g_i_*, *c_i_*, and *J_i_* (*i* = 1, 2, …, *n*) shown in Figure 2a are defined as the *i*th junction node connecting the transverse runner and the longitudinal runner, the gate location and the end of the cavity *J_i_* at the right side of the runner system, individually. To optimize the diameter of each runner, two important concepts resulting from the H-type runner system must be followed. The filling time and the pressure drop from any junction to each end of the cavity must be the same to achieve a balanced flow [8,16,17,18,19]. The four stages of the optimization methodology to construct the optimal runner system are described briefly as follows. (1) Find the final junction of the multi-cavity mold where several runners with different lengths diverge. (2) Select one of the runners as a benchmark and assign an initial filling time and the runner diameter. (3) Calculate the filling velocity and rheological property of the runners branching from the same junction node to determine their pressure drops. (4) Make all of the pressure drops of those runners the same as that of the benchmark runner by adjusting their diameters. Finally, those diameters are the optimal diameters which make the molten plastic filling occur in a balanced flow pattern.

In the filling stage of the injection process, molten plastic flow starts sequentially from sprue, runner, and gate and finally arrives at the mold cavity. Conversely, the optimization procedure starts from the final junction *k_n_* where the final branch is divided into two runners: $L_{n}$ and $L_{n}^{\prime }$, which should first be considered. Assume the diameter of both runners equally and its value is *ϕ*_0_. The runner $L_{n}$ is selected as a benchmark and an initial filling time of the molten plastic flowing in the runner *L_n_* is given by *t*_0_. Suppose the molten plastic is in a fully developed state and the filling velocity is steady. The filling velocity in the runner *L_n_* can be calculated by:(9)$$v_{L_{n}}=\frac{k_{n}g_{n}}{t_{0}}$$
where $k_{n}g_{n}$ represents the length of runner *L_n_* defined by the distance from *k_n_* to *g_n_*. The filling velocity $v_{J_{n}}$ at which the molten plastic flows into the cavity can also be determined based on the law of conservation of mass [18]. Since the filling velocity in the runner is known, the shear rate of the runner can be calculated by Equation (6) and its viscosity can be referred to material rheological data. Therefore, the pressure drop $\Delta P_{L_{n}}$ for the molten plastic filling in the runner *L_n_* can be calculated by Equation (8). Since the shape of the mold cavity is the same for the multi-cavity mold, the pressure drop $\Delta P_{J_{n}}$ of the cavity $J_{n}$ is also assumed to be proportional to $\Delta P_{L_{n}}$ and can be expressed by:(10)$$\Delta P_{J_{n}}=\Delta P_{L_{n}}{\left[\frac{Q_{L_{n}}\left(3s+1\right)}{sv_{J_{n}}A_{J_{n}}}\right]}^{s}$$
where $A_{J_{n}}$ is the cross-sectional area of the cavity and $Q_{L_{n}}$ is the volume flow rate in the runner *L_n_*. Similarly, the pressure drop $\Delta P_{L_{n}^{\prime }}$ of another runner $L_{n}^{\prime }$ and the pressure drop $\Delta P_{J_{n}^{\prime }}$ of the cavity $J_{n}^{\prime }$ can also be computed in accordance with the same procedure. Remember that the pressure drop in different runners diverging from the same junction should be the same. Therefore, the following criterion expressed by Equation (11) must exist for a balanced flow pattern.
(11)$$\begin{array}{c} \underset{︸}{\Delta P_{L_{n}}+\Delta P_{J_{n}}}\\ \mathrm{pressure}\;\mathrm{drop}:\;k_{n}\;\mathrm{to}\;c_{n} \end{array}=\begin{array}{c} \underset{︸}{\Delta P_{L_{n}^{\prime }}+\Delta P_{J_{n}^{\prime }}}\\ \mathrm{pressure}\;\mathrm{drop}:\;k_{n}\;\mathrm{to}\;c_{n}^{\prime } \end{array}$$

The lengths of both runners $L_{n}$ and $L_{n}^{\prime }$ from the same junction are different; therefore, the diameter of the runner $L_{n}^{\prime }$ should be reduced or enlarged to eliminate the difference of the pressure drops shown in Equation (11).

As the optimal diameters of runners $L_{n}$ and $L_{n}^{\prime }$ are determined, take the next junction *k_n_*_−1_ for optimization. When the molten plastic arrives at *k_n_*_−1_ and prepares for further filling advancement, the pressure drop of the molten plastic filling from *k_n_*_−1_ to *c_n_* and $c_{n}^{\prime }$ must be the same as that of the molten plastic filling from *k_n_*_−1_ to *c_n_*_−1_ and $c_{n-1}^{\prime }$. It means that the sum of the pressure drops of the molten plastic from *k_n_*_−1_ to cavities $J_{n-1}$ and $J_{n-1}^{\prime }$ must equal that from *k_n_*_−1_ to cavities $J_{n}$ and $J_{n}^{\prime }$, as well as the pressure drop in the runner *T_n_*:(12)$$\begin{array}{c} \underset{︸}{\Delta P_{L_{n-1}}+\Delta P_{J_{n-1}}+\Delta P_{L_{n-1}^{\prime }}+\Delta P_{J_{n-1}^{\prime }}}\\ \mathrm{pressure}\;\mathrm{drop}:\;k_{n-1}\;\mathrm{to}\;J_{n-1}\;\&\;J_{n-1}^{\prime } \end{array}\;\;\;\;\;\;\;\;\;\;\;=\begin{array}{c} \underset{︸}{\Delta P_{L_{n}}+\Delta P_{J_{n}}+\Delta P_{L_{n}^{\prime }}+\Delta P_{L_{n}^{\prime }}}\\ \mathrm{pressure}\;\mathrm{drop}:\;k_{n}\;\mathrm{to}\;J_{n}\;\&\;J_{n}^{\prime } \end{array}+\begin{array}{c} \underset{︸}{\Delta P_{T_{n}}}\\ \mathrm{pressure}\;\mathrm{drop}:\;k_{n-1}\;\mathrm{to}\;k_{n} \end{array}$$

Note that the runner $T_{n}$ is the bridge connecting *k_n_*_−1_ to *k_n_* and the pressure drop $\Delta P_{T_{n}}$ must be added in the pressure balancing relation of Equation (12). The filling velocity in the runner *T_n_* can be calculated based on the law of conservation of mass and expressed by:(13)$$v_{T_{n}}=\frac{1}{\emptyset_{T_{n}}^{2}}\left(\emptyset_{L_{n}}^{2}v_{L_{n}}+\emptyset_{L_{n}^{\prime }}^{2}v_{L_{n}^{\prime }}\right)$$
where $\emptyset_{T_{n}}$ is the diameter of the transverse runner *T_n_* which equals the average of both runners $L_{n}$ and $L_{n}^{\prime }$. As the filling velocity in the runner *T_n_* is known, the pressure drop $\Delta P_{T_{n}}$ can be calculated by Equation (8). Since the locations of the cavities $J_{n-1}$ and $J_{n-1}^{\prime }$ are nearer to the sprue than the cavities $J_{n}$ and $J_{n}^{\prime }$ the sum of the pressure drops when the molten plastic flow from $k_{n-1}$ to $J_{n-1}$ and $J_{n-1}^{\prime }$ must be less than the sum of the pressure drops when the molten plastic flow from $k_{n-1}$ to $J_{n}$ and $J_{n}^{\prime }$ plus the pressure drop $\Delta P_{T_{n}}$. To satisfy the relation defined by Equation (12), reduce the diameters of runners $L_{n-1}$ and $L_{n-1}^{\prime }$ until the pressure drops on both sides of Equation (12) are equal. As a result, the optimal diameters of the runners $L_{n-1}$ and $L_{n-1}^{\prime }$ are determined. Same calculation procedures can be implemented for the next junction and to decide the optimal diameters for the other runners.

## 4. Initial Conditions and Validation

Some initial conditions need to be addressed before the execution of the optimization methodology can be demonstrated. The materials used in the examples include polypropylene (Globalene, PP-6331, from LCY Chemical Corp., Taiwan) and acrylonitrile butadiene styrene (ABS, POLYLAC PA-757, from CHIMEI Corp., Taiwan). The recommended melt temperature for both resins is 180~220 °C for PP-6331 and 190~230 °C for PA-757, respectively. All of their rheological information of the viscosity with respect to shear rate can be referred to the Autodesk Moldflow Insight (Moldflow) software [16]. The optimal results calculated by the proposed method will be validated by Moldflow. Two results, the melt front advancement provides a graphical indication of how the molten plastic is filled in the mold, while the injection pressure confirms an improved balanced runner design numerically. A low injection pressure also implies that the residual stress of the molded product is less. The initial filling time of the molten plastic flow in the first runner will be estimated in accordance with the total volume of the molded product and the runner system with respect to a commercial injection machine capacity (FANUC, ROBOSHOTα-S130iB).

## 5. Results and Discussion

Three examples demonstrate how the proposed method is executed to design the runner system of the multi-cavity mold. The first example introduces a fishbone runner system of 16 mold cavities. Besides determining the diameters of the runner system, the melt temperature effect on the runner system design is also investigated. The design of an arbitrary runner layout of a multi-cavity mold for a balanced flow is implemented in the second example. The influence of the melt temperature and resin type on the runner design is illustrated in the third example.

### 5.1. The Optimal Runner Design of a 16-Cavity Mold with a Fishbone Layout

In general, the layout of the fishbone runner system and the cavity location is a symmetrical arrangement so that the molten plastic filling in the mold is a symmetrical condition. For the sake of saving computation cost, only a half layout of the runner system of a16-cavity mold is demonstrated, as shown in Figure 3. The nodes *k_i_*, *g_i_*, and *c_i_* represent the junction, gate, and the end of the *i*th cavity. The material used for this example is PP-6331 and the melt temperature is set at 220 °C. The viscosity with respect to shear rate of PP-6331 is referred to the Moldflow software [16]. The initial settings of all the runner diameters are 6.5 mm as well as the initial filling time in the runner *L*_4_, whose length is defined from *k*_4_ to *c*_4_ is 0.2 s. The initial filling time in the runner is estimated in accordance with the injection rate of injection machine (FANUC, ROBOSHOTα-S130iB).

The optimization methodology starts from the final junction *k*_4_; and then transfers to the junctions *k*_3_, *k*_2_, and *k*_1_ to determine each runner diameter sequentially. All of the rheological properties in the transverse and the longitudinal runners of the two runner system designs (the original and the improved) are calculated and listed in Table 1. The runner diameters of the original design are the same and the filling time at each runner *T*_1_~*T*_4_ is increased with respect to the distance from the sprue, signifying that the shear rates at the runners *T*_1_~*T*_4_ are quite different. Compared to the improved runner design, the runner diameters are optimized so that the filling time is similar and the difference of the shear rate at each runner is less. Furthermore, it implies that the viscosity of the molten plastic in the runner system is close so that a balanced flow pattern can be achieved. The same result can also be observed at runners *L*_1_~*L*_4_. The pressure drop of the runner provides another information on how a balanced flow pattern is achieved. An imbalanced flow of the runner system always causes much injection pressure to fill a mold. The validation of the melt front advancements is simulated by Moldflow. The original runner design exhibits an imbalanced flow result illustrated in Figure 4a,b. More balanced melt front advancements are achieved by using the optimal runner system, as shown in Figure 4c–d. In addition, the injection pressures of both cases shown in Figure 4e,f prove that the product molded by the optimal design can be implemented under relatively low pressure.

### 5.2. The Runner Design of an 8-Cavity Mold with an Arbitrary Runner Layout

In order to save mold space or reduce the mold cost, an arbitrary runner layout is usually seen in the family mold or in pilot run stage of the new product. There is an 8-cavity mold and only half of the runner system is illustrated in Figure 5. The original design of the all runner diameters is 6.5 mm. A reasonable filling time for the molten plastic to flow in the runners *T*_4_ and *L*_4_ is 0.27 s in accordance to the injection machine capacity. Since junction k4 is the connection of the runner *T*_4_ and the runner *L*_4_ with no branch, the diameters of the runners *T*_4_ and *L*_4_ are set equally at 8.0 mm. The junction *k*_3_ is the first node where two runners branch and begins to compute the diameter of the runner *L*_3_ in comparison with the runners *T*_4_ and *L*_4_. Note that the pressure drop of the cavity also needs to be considered in the optimization process. In accordance with the proposed method, the comparison of the original design and the optimal design of the runner system are listed in Table 2. The total runner volumes of both designs are similar. However, the optimal design provides a balanced flow and the required injection pressure is lower than that of the original runner system, an improvement of about 8.3%. The validation is also executed by Moldflow and the results are shown in Figure 6.

### 5.3. The Influences of Melt Temperature and Resin Type on the Runner Design of a Multi-Cavity Mold

The melt viscosity influences the pressure required for the molten plastic filling the cavity. It represents the resistance from friction in the runner system and the cavity on the filling stage. Three melt temperatures: 180, 200, and 220 °C were selected for the optimization of the fishbone runner system. The layout of the runner system and the mold cavity is a 16-cavity mold demonstrated in the previous example. After the optimization calculation, the diameters of the runner system, the injection pressure and the runner volume with respect to three melt temperatures are listed in Table 3. The adjacent runner diameter change becomes apparent at a lower melt temperature since the pressure drop of the molten plastic at a lower melt temperature increases because the melt viscosity rises. To achieve a balanced flow, the runner diameter should vary much more to eliminate the effect of high viscosity.

Similarly, the melt flow index (MFI) of two resin types: PP-6331 and ABS PA-7533, are 14 and 7 g/10 min, respectively. It means that the flowability of PP-6331 is easier than that of ABS PA-7533 near two times. Both viscosity differences are much larger so that the effect of resin type is significantly larger than that of the melt temperature. The results can be seen in Table 4.

## 6. Conclusions

For a multi-cavity mold design, the fishbone runner system is welcomed since the waste of the runner is less. However, an imbalanced flow behavior makes product quality molded by the fishbone runner system low. The proposed method enables a balanced flow pattern to be easily produced by the fishbone runner system. The results indicate that a more balanced flow of injection molding provides a lower injection pressure, which improves the internal residual stress of the molded product. An analytical solution enables us to make a parametric check during the design process, whereas the commercial software cannot be checked when only numerical values are used. The results indicate that the runner diameter change of the molten plastic at relatively high viscosity becomes apparent so as to achieve a more balanced flow pattern. The proposed method is not meant to replace the commercial software. In contrast, this method shows its great benefit when a new runner design project is launched in the initial design stage and then cooperates with the commercial software for further modifications.

## Figures and Tables

**Figure 1 polymers-14-05442-f001:**
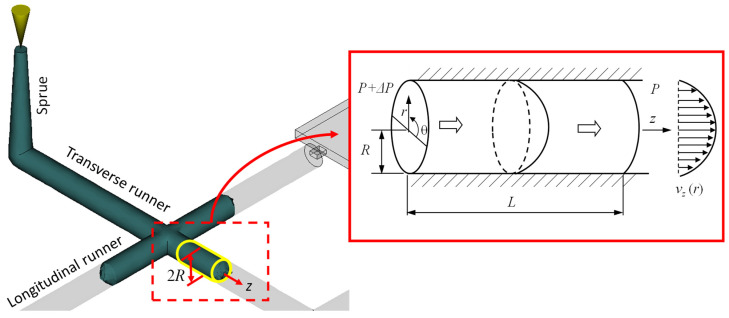
The flow behavior of the molten plastic in the circular runner.

**Figure 2 polymers-14-05442-f002:**
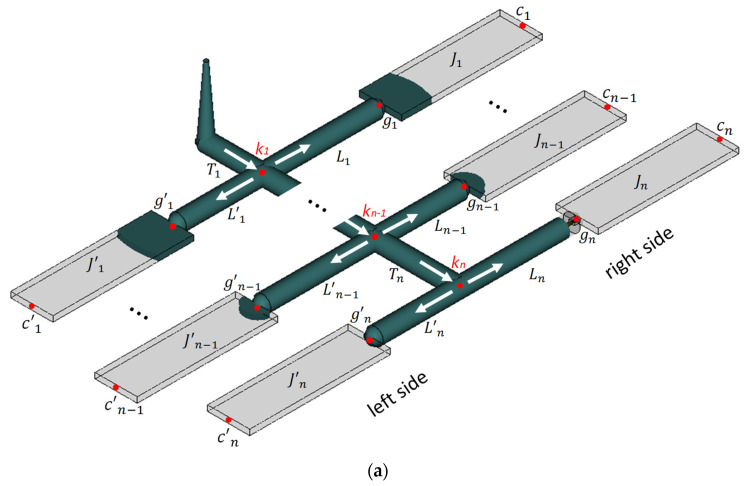
(**a**) A half model of the runner system of a 4*n*-cavity mold illustration, (**b**) the pressure drop balance at the final junction $k_{n}$, (**c**) the pressure drop balance at the final junction $k_{n-1}$.

**Figure 3 polymers-14-05442-f003:**
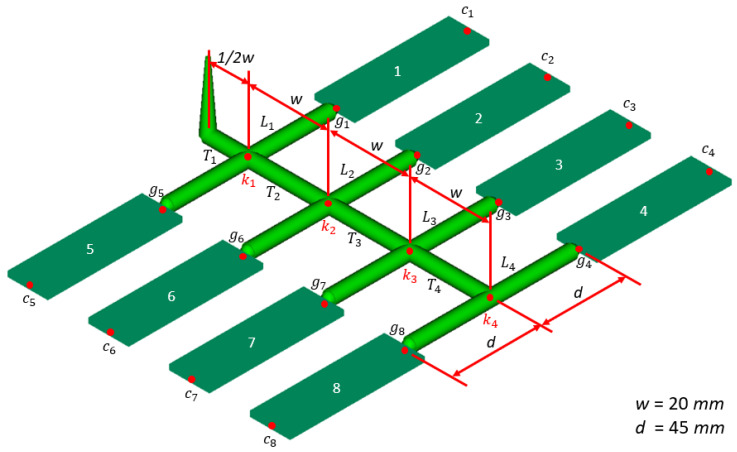
A half layout of the fishbone runner of a16-cavity mold.

**Figure 4 polymers-14-05442-f004:**
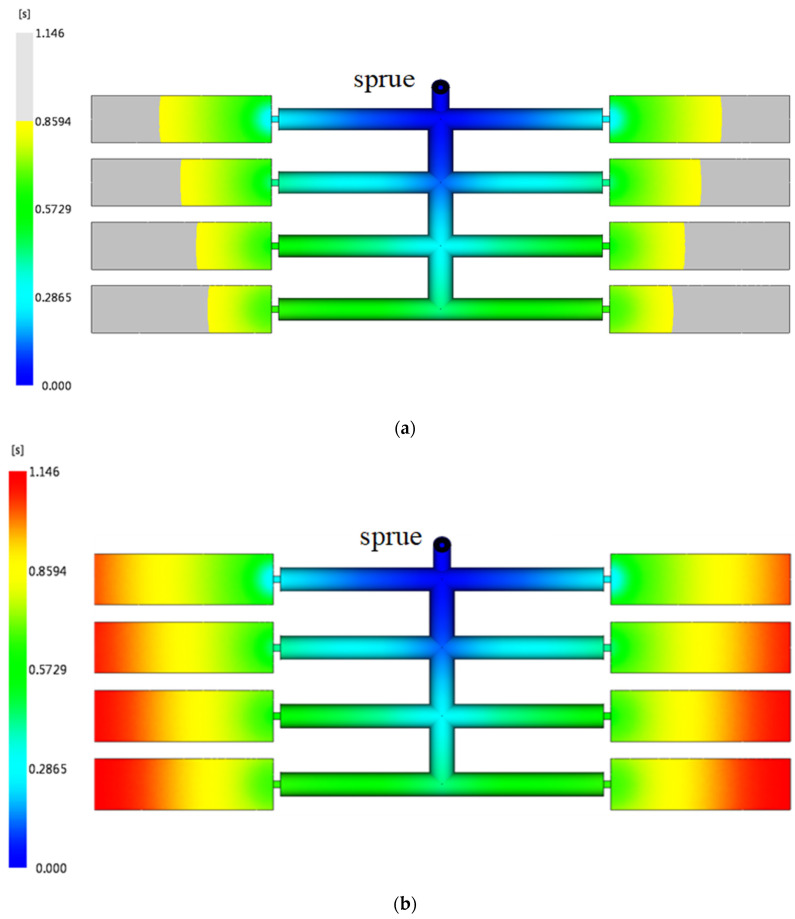
The comparison of an imbalanced and a balanced runner design of 16-cavity mold (a half model): (**a**) 75% melt fill of an imbalanced runner system, (**b**) 100% fill of an imbalanced runner system, (**c**) 75% fill of a balanced runner system, (**d**) 100% fill of a balanced runner system, (**e**) injection pressure of an imbalanced runner design, and (**f**) injection pressure of a balanced runner design.

**Figure 5 polymers-14-05442-f005:**
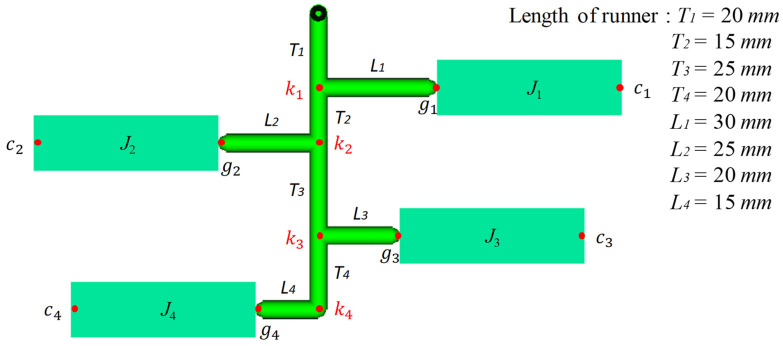
Illustration of 1/2 model of 8-cavity mold with an arbitrary runner layout.

**Figure 6 polymers-14-05442-f006:**
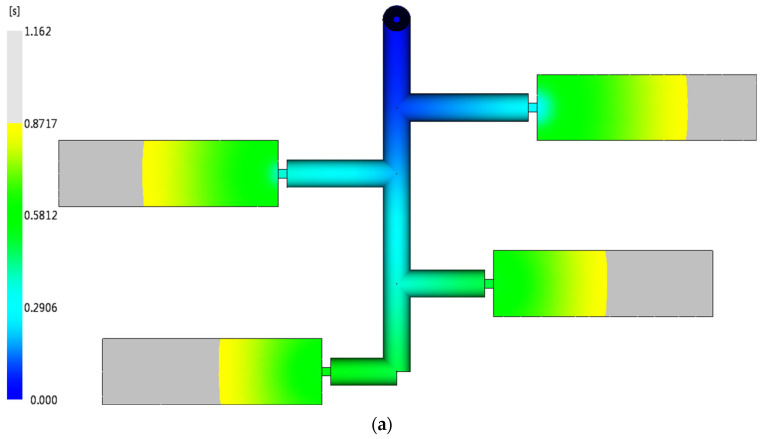
The comparison of an imbalanced and a balanced runner design of 8-cavity mold (a half model): (**a**) 75% melt fill of an imbalanced runner system, (**b**) 100% fill of an imbalanced runner system, (**c**) 75% fill of a balanced runner system, (**d**) 100% fill of a balanced runner system, (**e**) injection pressure of an imbalanced runner design, and (**f**) injection pressure of a balanced runner design.

**Table 1 polymers-14-05442-t001:** Rheological properties at each runner and the optimal diameters.

	Filling Time, s		Shear Rate, 1/s		Viscosity, Pa·s		Pressure Drop, MPa		Optimal Diameter, mm
	Original	Improved	Original	Improved	Original	Improved	Original	Improved	
*T* _1_	0.06	0.06	2.26	2.26	213	213	1.8	1.8	5.0
*T* _2_	0.15	0.14	2.14	2.15	237	263	1.7	2.2	6.3
*T* _3_	0.17	0.11	0.58	1.83	415	230	2.5	1.9	7.4
*T* _4_	0.20	0.04	0.22	1.95	561	215	2.8	1.7	8.8
*L* _1_	0.08	0.13	2.07	1.23	245	398	1.6	2.1	5.0
*L* _2_	0.09	0.11	1.75	1.42	263	345	2.2	1.9	6.2
*L* _3_	0.11	0.08	1.53	2.05	358	310	2.5	1.8	7.4
*L* _4_	0.19	0.07	0.18	2.10	653	320	3.7	2.3	8.8

**Table 2 polymers-14-05442-t002:** The runner diameter, volume, and injection pressure of the 8-cavity mold (a half model) before and after optimization.

	Fishbone Runner System								Runner Volume mm^3^	Injection Pressure MPa
	Transverse Runner *T_i_*				Longitudinal Runner *L_i_*					
	*i* = 1, 2, …, 4				*i* = 1, 2, …, 4					
Runner	$T_{1}$	$T_{2}$	$T_{3}$	$T_{4}$	$L_{1}$	$L_{2}$	$L_{3}$	$L_{4}$	-	-
Runner Length, mm	20	15	25	20	30	25	20	15	-	-
Original diameter, mm	6.5	6.5	6.5	6.5	6.5	6.5	6.5	6.5	5345	15.39
Optimal diameter, mm	5	6	7	8	4.65	5.25	6.75	8	5305	14.21

**Table 3 polymers-14-05442-t003:** Optimal diameters of the runner system and injection pressure at different melt temperature.

	Fishbone Runner System								Injection Pressure MPa	Runner Volume mm^3^
	Transverse Runner *T_i_*				Longitudinal Runner *L_i_*					
	*i* = 1, 2, …, 4				*i* = 1, 2, …, 4					
Runner	$T_{1}$	$T_{2}$	$T_{3}$	$T_{4}$	$L_{1}$	$L_{2}$	$L_{3}$	$L_{4}$	-	-
Runner Length, mm	10	20	20	20	45	45	45	45	-	-
220 °C, mm	5.0	6.3	7.4	8.8	5.0	6.2	7.4	8.8	16.1	16,576
200 °C, mm	5.0	6.1	7.5	9.0	5.0	6.2	7.4	8.9	18.5	16,720
180 °C, mm	5.0	5.9	7.7	9.3	5.1	6.5	7.3	9.3	21.5	16,854

**Table 4 polymers-14-05442-t004:** Optimal diameters of the runner system and injection pressure with respect to different resin type.

	Fishbone Runner System								Injection Pressure MPa	Runner Volume mm^3^
	Transverse Runner *T_i_*				Longitudinal Runner *L_i_*					
	*i* = 1, 2, …, 4				*i* = 1, 2, …, 4					
Runner	$T_{1}$	$T_{2}$	$T_{3}$	$T_{4}$	$L_{1}$	$L_{2}$	$L_{3}$	$L_{4}$	-	-
Runner Length, mm	10	20	20	20	45	45	45	45	-	-
ABS PA-757	5.0	6.5	8.2	9.5	5.0	6.2	7.4	8.8	65.7	16,576
PP 6331	5.0	6.3	7.4	8.5	6.8	7.3	7.4	7.5	53.2	17,674

## Data Availability

The data presented in this study are available on request from the corresponding author.

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
