# Peer review of "A Semi-Analytical Method for Designing a Runner System of a Multi-Cavity Mold for Injection Molding"

_polymers, 2022, doi:10.3390/polym14245442_

Round 1

Reviewer 1 Report

The topic taken up in the article is interesting and important, useful for designers of multi-cavity injection molds and manufacturers of precise elements made of polymeric materials. However, the article should be prepared more carefully, taking into account the comments made in the text of the work.

1. All symbols in equations should be explained.

2. Equation 11 is hard to read.

3. Table 1: If the filling time of individual channels is given in the table, e.g. material flow time from point k1 to k2, from k2 to k3, etc.? If so, why the filling time of the shorter T1 channel is longer than, e.g. of the T3 channel?

4. There are no rheological properties of the material in table 2. Channel lengths are different from those shown in Figure 5. Are these data for a 16-cavity or 8-cavity mold?

5. Other comments are in the text of the work.

Author Response

Dear reviewer,

The authors would like to thank the reviewer for his/her efforts and valuable suggestions. The reviewer’s comments are carefully replied  and the point-to-point reply to the comments are attached. Please see the attachement.

Best regards,

Reviewer 2 Report

This manuscript reported a semi-analytical method for designing a runner system of a  multi-cavity mold for injection molding with 3 examples provided. Although the manuscript displayed significance in practical processing, I cannot recommend its publication in current state and the suggestion are listed hereinafter.

1.      In the 2nd paragraph of section 5.1, authors list tons of data in Table1 without detailed discussions. Every data should support a specific conclusion, and the calculation procedure of the optimal size needs to be explained, which is the core of this study.

2.      Why the length of runner in Table2 is different from that on the right side of Fig5?

3.      There are lots of typo and format mistakes through the manuscript.

ü  The expression “ki, gi and ki” in line 165 is confusing. Do you mean “ki gi and ci”?

ü  The subtitle 5.1, 5.2 and 5.3 in the “3. Results and discussion” is wrong.

ü  The size in line 259 is different from others.

ü  There are many inconsistencies in the accuracy of the data in the article, such as 8mm in line 295 and 8.0mm in line 298.

ü  Why the last two row of Table2 are both original diameter?

ü  No scale illustration on the left of Fig6 (a)

ü  Typo mistakes of oC in Line 324

Author Response

(The authors gave the same response as above.)

Round 2

Reviewer 1 Report

All my comments have been taken into account in the new version of the manuscript, accept for one on the top of page 6: "...??? is the cross sectional area of the cavity..."

Reviewer 2 Report

All my concerns have been addressed and the manuscript can be accepted in current state.